# Major Adverse Cardiovascular Events: The Importance of Serum Levels and Haplotypes of the Anti-Inflammatory Cytokine Interleukin 10

**DOI:** 10.3390/biom14080979

**Published:** 2024-08-09

**Authors:** Susanne Schulz, Leonie Reuter, Alexander Navarrete Santos, Kerstin Bitter, Selina Rehm, Axel Schlitt, Stefan Reichert

**Affiliations:** 1Department of Operative Dentistry and Periodontology, Medical Faculty, Martin-Luther-University Halle-Wittenberg, 06108 Halle, Germany; 2Center for Medical Basic Research, Medical Faculty, Martin-Luther-University Halle-Wittenberg, 06120 Halle, Germany; 3Department of Cardiology, Paracelsus-Harz-Clinic Bad Suderode, 06485 Quedlinburg, Germany; axel.schlitt@pkd.de; 4Department of Medicine III, Medical Faculty, Martin-Luther-University Halle-Wittenberg, 06120 Halle, Germany

**Keywords:** interleukin 10, serum level, haplotype, cardiovascular disease, prognostic factor

## Abstract

Background: Cardiovascular diseases (CVDs) represent major medical and socio-economic challenges worldwide. There is substantial evidence that CVD is closely linked to inflammatory changes triggered by a complex cytokine network. In this context, interleukin 10 (IL-10) plays an important role as a pleiotropic cytokine with an anti-inflammatory capacity. In this study (a substudy of ClinTrials.gov, identifier: NCT01045070), the prognostic relevance of IL-10 levels and IL-10 haplotypes (rs1800896/rs1800871/rs1800872) was assessed regarding adverse cardiovascular outcomes (combined endpoint: myocardial infarction, stroke/transient ischemic attack, cardiac death and death according to stroke) within a 10-year follow-up. Patients and methods: At baseline, 1002 in-patients with CVD were enrolled. Serum levels of IL-10 were evaluated utilizing flow cytometry (BD™ Cytometric Bead Array). Haplotype analyses were carried out by polymerase chain reactions with sequence-specific primers (PCR-SSP). Results: In a survival analysis, IL-10 haplotypes were not proven to be cardiovascular prognostic factors in a 10-year follow-up (Breslow test: *p* = 0.423). However, a higher IL-10 level was associated with adverse cardiovascular outcomes (Breslow test: *p* = 0.047). A survival analysis considering adjusted hazard ratios (HRs) could not confirm this correlation (Cox regression: adjusted HR = 1.26, *p* = 0.168). Conclusion: In the present study, an elevated IL-10 level but not IL-10 haplotypes was linked to adverse cardiovascular outcomes (10-year follow-up) in a cohort of CVD patients.

## 1. Introduction

The anti-inflammatory cytokine IL-10 is secreted by macrophages, B cells and T helper cells in response to systemic inflammation. IL-10 has been shown to suppress antigen presentation, the activity of dendritic cells and the proliferation of T cells, monocytes and macrophages. It has a negative effect on the regulation of the production of pro-inflammatory cytokines [1]. Due to its anti-inflammatory properties, it is recognized as protecting people from the development and progression of cardiovascular alterations [2,3]. In fact, results from animal studies show that IL-10 prevents the formation and improves the stability of atherosclerotic plaques, in addition to supporting a reduction in lesion size [4,5,6]. IL-10 exerts its protective cardiovascular effect by suppressing inflammation, oxidative stress and endothelial monocyte adhesion [7]. Consistent with this, in a small-scale clinical study of patients with metabolic syndrome (n = 90), higher IL-10 levels were indeed associated with a lower prevalence of severe coronary heart disease [8]. Furthermore, a reduced serum IL-10 level is recognized to be a marker of plaque instability and a predictor of adverse outcomes after acute coronary syndrome [9,10]. However, other studies have failed to show a corresponding association [2], and some have even showed an inverse dependence [11,12]. These very contradictory clinical results suggest that there is variability in the relevance of IL-10 according to the clinical issue.

As could be observed, IL-10 production is inter alia regulated at the transcriptional level [13]. Single nucleotide polymorphisms (SNPs) in the promoter of the gene (rs1800896, rs1800871, rs1800872) play a decisive role in this respect [14]. Using luciferase reporter genes and electrophoretic mobility shift assays, the influence of individual nucleotide polymorphisms of the promoter on IL-10 transcriptional expression and binding to nuclear factors was investigated [14,15]. It has been proven that there are transcriptional variations between the haplotypes (ATA, ACC, GCC of rs1800896, rs1800871, rs1800872) and differences in IL-10 production under LPS stimulation [15]. In these studies, the ATA haplotype was associated with lower transcriptional activity and reduced IL-10 expression after LPS stimulation [15].

Due to the genetic influence on IL-10 expression, it is conceivable that the genetic constellation is also associated with the occurrence and prognosis of cardiovascular events. However, in a meta-analysis evaluating the impact of SNPs rs1800896, rs1800871 and rs1800872 on CVD, rs1800871 was associated with modified cardiovascular risk [16]. In a further meta-analysis addressing this issue, it was demonstrated that the SNP rs1800896 is associated with early or severe expression of CVD [17]. A meta-analysis conducted in East Asia confirmed the link between SNPs (rs1800871, rs1800896) and the occurrence of CVD [18]. However, little is known about the predictive cardiovascular value of IL-10 genetic variants. A small-scale study (n = 48) was conducted with patients with peripheral arterial disease to investigate whether the IL-10 haplotypes (rs1800896, rs1800871, rs1800872) can be utilized as prognostic factors for cardiovascular outcomes [19]. Herein, the ATA haplotype was associated with acute postoperative cardiovascular events in a short follow-up period of 30 days [19].

The objective of this longitudinal study was to assess the importance of IL-10 serum levels as well as IL-10 SNPs and haplotypes (rs1800896, rs1800871, rs1800872) in cardiovascular outcomes in a cohort of CVD patients. During a follow-up period of 10 years, the recurrence of adverse cardiovascular events, namely cardiovascular death, death from stroke, stroke/transient ischemic attack (TIA) and myocardial infraction (MI), was evaluated.

## 2. Materials and Methods

### 2.1. Study Population

This study represents a substudy of the longitudinal cohort study “Periodontitis and Its Microbiological Agents as Prognostic Factors in Patients with Coronary Heart Disease” (ClinicalTrials.gov identifier: NCT0104507).

Specific details of the clinical assessment of the patients were described in depth in Reichert et al. (2016) [20]. Explained briefly, 1002 patients suffering with CVD who underwent coronary angiography were prospectively included from October 2009 to February 2011.

The criteria for inclusion were an age of ≥18, ≥50% stenosis of a main coronary artery proven by current or previous coronary artery bypass surgery (CABG) or by coronary angiography and a current or previous percutaneous coronary intervention (PCI).

The criteria for exclusion were antibiotic treatment (within the last 3 months), non-surgical (scaling and root planing) or surgical periodontitis therapy (within the last 6 months), pregnancy, an inability to give written consent or the current abuse of alcohol or drugs.

The clinical assessment was conducted the day after hospitalization. Fasting blood withdrawal was conducted between 7 a.m. and 8 a.m. Serum parameters were determined in the Central Laboratory of the Medical Faculty of the Martin-Luther-University Halle-Wittenberg using standard procedures. Demographic and anamnestic variables (age, gender, smoking behavior, body mass index) and the patients’ disease history (e.g., myocardial infarction, stroke, TIA, hypertension, peripheral arterial disease, diabetes mellitus, dyslipoproteinemia) were collected.

In the 10-year follow-up (November 2020–January 2023), the predefined combined CV endpoint (MI, CV death, stroke/TIA, death from stroke) was requested. A standardized questionnaire was sent to patients by post. After unsuccessful attempts (twice after 8 weeks), the patients or their relatives were contacted by telephone. In the case of a patient’s death, the attending physician was consulted. If necessary, the registry offices were asked about the address or date of death.

### 2.2. Interleukin 10 Analysis

Genetic analyses of genotypes, alleles and haplotypes of rs1800896/rs1800871/rs1800872 were conducted by applying PCR-SSP (sequence specific oligonucleotides). For this purpose, the CYTOKINE Genotyping array CTS-PCR-SSP kit (Collaborative Transplant Study, Department of Transplantation Immunology of the University Clinic of Heidelberg, Germany) was used in accordance with the manufacturer’s instructions.

The serum concentration of IL-10 was assessed at the center for Medical Basic Research of the medical faculty (Martin-Luther-University Halle-Wittenberg). Flow cytometry was performed using LSR II Fortessa (BD Bioscience, Heidelberg, Germany). The BD™ Cytometric Bead Array for human IL-10 (BD Bioscience, Heidelberg, Germany) was applied according to the manufacturer’s instructions.

### 2.3. Statistical Analyses

Statistical analyses were performed with commercially available software SPSS 25.0 (SPSS Inc., Chicago, IL, USA). *p* values ≤ 0.05 were assessed as significant. Testing for normal distribution was conducted by the Kolmogorov–Smirnov test and the Shapiro–Wilk test (continuous data: metric clinical, serological data and demographic data). All metric variables were non-normally distributed and therefore documented as the median and 25th/75th interquartiles. Statistics were assessed using the Mann–Whitney U test or the Wilcoxon test. Categorical variables were reported as percentages. The Chi-squared test (including Fisher’s exact test: expected values < 5) was applied for the statistical analyses.

In survival analyses, using Kaplan–Meier plots, the Breslow rank test (univariate analyses) and Cox regression (adjusted hazard ratios; HRs), the influence of IL-10 levels and genetic variants of the IL-10 gene rs1800896/rs1800871/rs1800872 on CV outcomes was assessed. In order to generate Kaplan–Meier plots, metric IL-10 values were dichotomized with respect to the calculated median (≤1.5 pg/mL vs. >1.5 pg/mL). Adjusted HRs were generated for IL-10 levels with Cox regression with respect to known risk factors for CVD.

## 3. Results

### 3.1. Characterization of CV Patients

A total of 1500 patients with CVD were screened for eligibility. Among the patients, 1002 matched the inclusion criteria and were consequently enrolled. In the baseline investigation, 942 patients were included. For a genetic analysis of IL-10 haplotypes, 940 patients were available, and for protein biochemical analyses, 421 patients were available. After the 10-year follow-up, the combined endpoint (cardiovascular deaths, deaths from stroke, MIs and strokes/TIAs) was determined (Figure 1: flowchart of study design).

### 3.2. Baseline Investigation: Clinical and Biochemical Characterization of CV Patients

At baseline, the median age of the patients was 69.1 years (25/75 IQR: 60.4/74.7). The percentage of women was 26.7%, and 10.6% of the patients were current smokers. Among the patients, 34.5% suffered from diabetes mellitus, 87.6% from hypertension, 59.2% from hyperlipoproteinemia, 9.7% from peripheral artery disease, 38.9% had already experienced an MI and 12.5% had experienced a stroke/TIA. Biochemical parameters were also recorded. The patient cohort showed increased values for C-reactive protein (8.6 mg/L (25/75 IQR: 3.7/32.9)) compared to the reference range (<5 mg/L). The values measured for HDL cholesterol (1.0 mmol/L (25/75 IQR: 0.8/1.2)) in CVD patients were lower than the reference values (>1.5 mmol/L). All other parameters determined as part of this study were within the internal laboratory reference range (IL-6: 7.8 pg/mL, 25/75 IQR: 3.8/15.9; leukocytes: 7.8 Gpt/L, 25/75 IQR: 6.4/9.7; creatinine: 87.0 mmol/L, 25/75 IQR: 73/109; total cholesterol: 4.3 mmol/L, 25/75 IQR: 3.7/6.3; LDL cholesterol: 2.5 mmol/L, 25/75 IQR: 2.0/3.3; triglycerides: 1.4 mmol/L, 25/75 IQR: 1.0/1.9).

Furthermore, IL-10 serum levels and genetic variants rs1800896/rs1800871/rs1800872 were evaluated (Figure 2). It could be shown that IL-10 serum levels were not correlated with IL-10 SNPs or haplotypes.

The mean follow-up time was 10.2 years (±2.6), and the number of major adverse cardiovascular events analyzed was 83 (8.9%).

### 3.3. Ten-Year Follow-Up: IL-10 Serum Level as Prognostic Marker for Further CV Events

In order to evaluate the influence of IL-10 serum level on cardiovascular outcomes, IL-10 levels were dichotomized. The median was used to allocate the groups (<1.5 vs. ≥1.5 pg/mL). An increased IL-10 serum level was revealed to be associated with further CV events on the Kaplan–Meier survival curve (Breslow test: *p* = 0.047; hazard ratio: 1.3; Figure 3).

In the Cox regression analysis, the adjusted hazard ratios were evaluated, taking into account established CV risk markers. Covariates of male gender, age, BMI, smoking, diabetes mellitus, hypertension and dyslipoproteinemia were considered (Table 1). In this complex model, an elevated IL-10 level could not be confirmed as an independent prognostic factor for a further cardiovascular event (HR: 1.26, 95% CI: 0.91–1.76). However, it was shown that older age and the presence of diabetes mellitus were significantly associated with an adverse CV outcome. 

Due to the high percentage of patients with IL-10 expression below the detection limit, the anamnestic and clinical data of the patients were analyzed in relation to the IL-10 levels. The results are listed in the Appendix A. None of the reported data differed significantly between the two patient groups except for CRP. Patients with IL-10 expression above the detection limit were characterized by an increased CRP level (Mann–Whitney U test: *p* = 0.001). In a bivariate correlation analysis, it was shown that the levels of CRP and IL-10 are positively linked (Spearman Rho: 0.113; *p* > 0.001).

In a further analysis, the 25 and 75 interquartile range was used as a separating parameter. Here, too, it is clear that patients with an increased expression of IL-10 showed an adverse cardiovascular outcome (Breslow test: *p* = 0.047; hazard ratio: 1.6; Figure 4). Again, an increased IL-10 level could not be confirmed as an independent prognostic factor in the multivariate Cox regression analysis (*p* = 0.113; HR: 1.45; 95% CI: 0.92–2.3).

### 3.4. Ten-Year Follow-Up: Genetic Variants of IL-10 Gene as Potential Prognostic CV Markers

The SNPs rs1800896, rs1800871 and rs1800872, as well as the corresponding haplotypes, were evaluated regarding their impact on adverse cardiovascular outcomes. In Kaplan–Meier statistics considering survival time, all SNPs were tested on the incidence of cardiovascular deaths, deaths from stroke, MIs and strokes/TIAs as the combined endpoint. However, no significant genetic influence on cardiovascular outcomes could be described (Figure 5).

## 4. Discussion

The present study showed that an elevated IL-10 level but not the genetic background of the IL-10 gene (alleles, genotypes, and haplotypes of rs1800896/rs1800871/rs1800872) was associated with an adverse cardiovascular prognosis in a cohort of patients with angiographically proven CVD. Among the participants, 1002 patients were enrolled, observed over a period of 10 years (378.3 ± 214.8 weeks) and evaluated for further cardiovascular events. To the best of our knowledge, this study is the first to investigate the significance of both the genetic background (rs1800896/rs1800871/rs1800872) and the serum level of the anti-inflammatory cytokine IL-10 in the occurrence of a future cardiovascular event.

### 4.1. IL-10 Serum Level as Prognostic Marker for Further CV Events

As an anti-inflammatory cytokine, interleukin 10 plays an important role in systemic inflammation and associated processes, such as cardiovascular diseases. A significant contribution has been ascribed to IL-10 in preventing the formation of atherosclerotic plaques and as a modulator of plaque instability [2,21]. In the present study, an increase in IL-10 baseline levels has been linked with an increased risk of suffering future cardiovascular events in a 10-year follow-up (*p* = 0.047; HR = 1.3) in a cohort of patients with angiographically proven coronary stenosis. This result is in contrast to the anti-inflammatory and anti-atherosclerotic properties associated with IL-10, which suggest that reduced IL-10 levels are linked with increased cardiovascular risk.

Indeed, IL-10 levels were analyzed in various studies in relation to cardiovascular outcomes [9,22]. In line with the hypothesis concerning its anti-atherogenic properties, an increased IL-10 level has been associated with a better prognosis after cardiovascular disease [10,22]. However, epidemiological evidence does not consistently support this link. It could be shown that the predictive power of the IL-10 level for cardiovascular events is strongly dependent on the study design. In a substudy of the Multi-Ethnic Study of Atherosclerosis (MESA; n = 930, age 48–90 years), participants were followed for an average of 10.2 years, and cardiovascular outcomes were assessed in relation to their IL-10 levels [2]. In this study, no predictive potential of the IL-10 level for cardiovascular events could be shown. However, in contrast to the present study, only patients without previous cardiovascular disease were examined in the MESA study.

On the other hand, in the PROSPER (Prospective Study of Pravastatin in the Elderly at Risk, age: 70–82 years) cohort, elevated levels of IL-10 were associated with a slight increase in adverse cardiovascular events over a follow-up period of 3.2 years [23]. However, the association was more strongly pronounced in elderly individuals without a history of CVD compared to individuals with a pre-existing cardiovascular condition (OR: 1.42 vs. 1.04). Also, this study can only be compared with the presented study to a limited extent, as the age at which the patients were included differed greatly.

Furthermore, in the estrogen replacement and atherosclerosis (ERA) study, an elevated baseline level of IL-10 was correlated with cardiovascular outcomes in post-menopausal women with angiographically proven coronary artery disease (n = 309, follow-up of 3.2 years) [12]. In comparison to the present study, only women were examined in the ERA study.

In summary, it can be assumed that the high variability between the cohorts in terms of age [23], gender [12] and pre-existing cardiovascular disease probably contributes to the lack of consistent results. Although the IL-10 level can be a predictive cardiovascular marker in certain constellations, its usefulness for predicting cardiovascular risk in the general population remains open to discussion.

### 4.2. Genetic Markers of IL-10 as Prognostic Markers for Further CV Events

IL-10 expression is partly determined by genetic predisposition. Although our study failed to prove a link between genetic background (rs1800896/rs1800871/rs1800872) and IL-10 expression (Figure 2), other studies have confirmed this association after lipopolysaccharide stimulation [15,24]. In these studies, the ATA haplotype was accompanied by low IL-10 expression in patients with juvenile rheumatoid arthritis [15] and patients undergoing elective cardiopulmonary bypass surgery [24]. It is possible that haplotype-dependent expression patterns were not evident in the present study as no inflammatory response was examined.

Nevertheless, since studies have shown a genetic influence on the expression of IL-10 levels, it is conceivable that genetic characteristics also contribute to the clinical impact of IL-10. However, different clinical case–control studies conducted to analyze this objective have revealed controversial results. In a meta-analysis evaluating case–control studies regarding the impact of rs1800896, rs1880871 and rs180872 on CVD, only rs1800871 was linked to the occurrence of cardiovascular disease [16]. Carriers of the T allele of the SNPs showed a 10% lower risk of CVD [16]. This is in line with the results obtained in an Asian population [18,25]. In addition, the Asian meta-analysis by Zheng et al. also revealed the SNP rs1800896 to be an important marker for CVD [18]. In a further meta-analysis, the relevance of the A allele of rs1800896 in the incidence of early-onset or severe CVD was emphasized [17].

However, the aim of the present study was to evaluate the influence of genetic variants in the IL-10 gene on future cardiovascular events. Here, we could not prove an important impact of the SNPs rs1800896, rs1800871 and rs1800872 on further myocardial infarction, cardiovascular death, stroke/transient ischemic attack or death from stroke in the 10-year follow-up (Figure 4). However, so far, there is little information addressing this topic. One study investigated the prognostic relevance of the genetic background of IL-10 in a cohort of patients suffering from peripheral artery disease [19]. In this study, the ATA haplotype of rs1800896/rs1800871/rs1800872 was identified as a risk factor for postoperative cardiovascular events. However, it is critical to note that only 48 patients were included in this study, and the follow-up period was only 30 days. For these reasons, the results cannot be compared with the results of the present study. In addition, because of differences in the inclusion criteria, no consistent conclusions can be drawn on the prognostic impact of the SNPs rs1800896, rs1800871 and rs1800872 and haplotypes on future cardiovascular events.

### 4.3. Limitations of This Study

This study was performed according to the design of a longitudinal cohort study. Longitudinal studies can provide a comprehensive research approach that allows for an understanding of the level and direction of alterations over time. However, there are some biases associated with this type of study design [26]. The causality of the results cannot be assumed due to the study design and the multifactorial nature of the disease. Furthermore, this study was conducted with a limited population in a single university hospital, which calls into question the generalizability and applicability of these data to different groups. Therefore, the present results are valid for patients with angiographically proven cardiovascular stenosis in Saxony-Anhalt, Central Germany. The results cannot be generalized to the overall population or other patient cohorts.

## 5. Conclusions

The present study showed that an increased IL-10 level is associated with an adverse cardiovascular outcome (10-year follow-up) in a cohort of CVD patients. However, this influence could not be confirmed when other cardiovascular risk factors were taken into account. Also, genetic variants in the IL-10 gene (rs1800896, rs1800871, rs1800872) could not be evaluated as prognostic cardiovascular factors. Therefore, based on current knowledge, it would be premature to integrate the IL-10 level or genetic characteristics of the IL-10 gene into the cardiovascular prognosis profile. Further clinical investigations are necessary to confirm possible links.

## Figures and Tables

**Figure 1 biomolecules-14-00979-f001:**
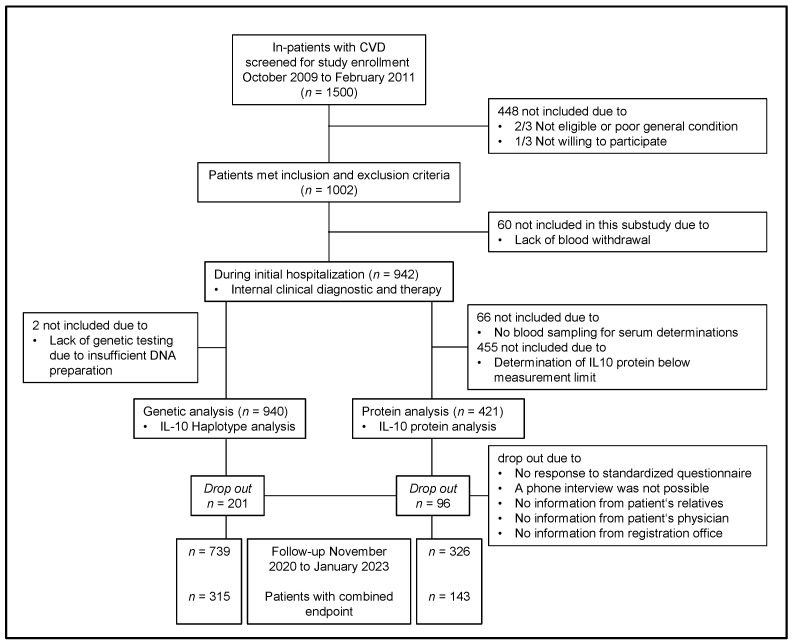
Study design.

**Figure 2 biomolecules-14-00979-f002:**
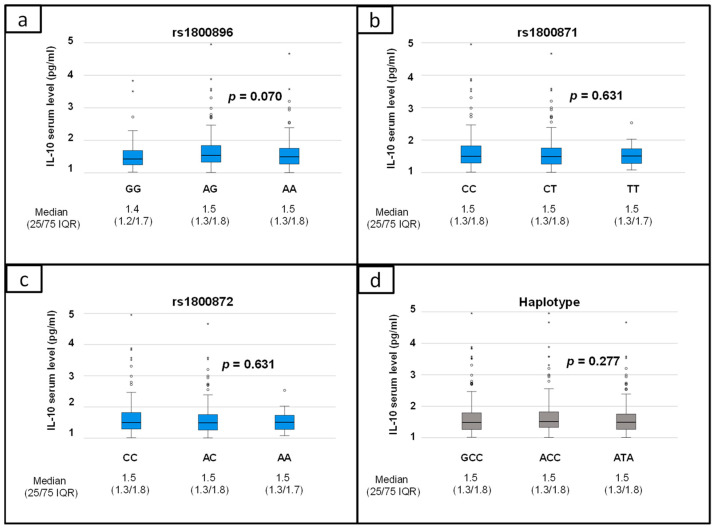
Relationship between IL-10 serum levels and genetic variants in IL-10 gene: (**a**) rs1800896; (**b**) rs1800871; (**c**) rs1800872; (**d**) haplotypes of rs1800896/rs1800871/rs1800872 (IQR: interquartile range).

**Figure 3 biomolecules-14-00979-f003:**
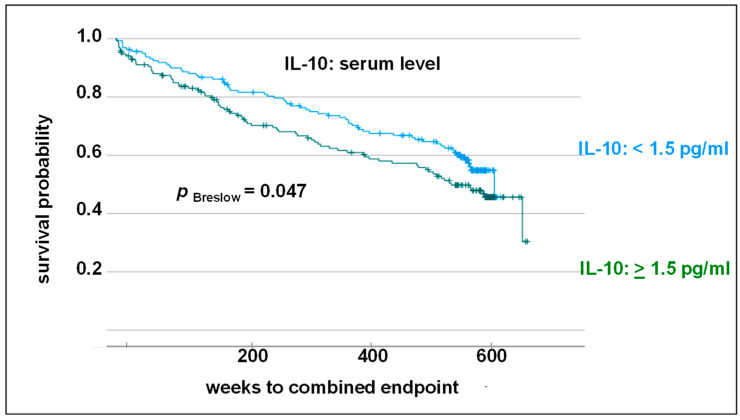
Kaplan–Meier survival curve and Breslow test. Impact of IL-10 level (<1.5 pg/mL vs. ≥1.5 pg/mL) on combined endpoint (myocardial infarction, cardiovascular death, stroke/transient ischemic attack and death from stroke) in 10-year follow-up.

**Figure 4 biomolecules-14-00979-f004:**
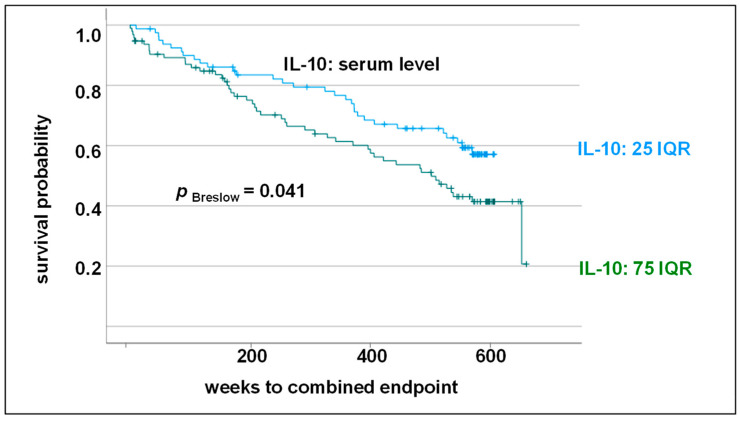
Kaplan–Meier survival curve and Breslow test. Impact of IL-10 level (IQR25: <0.45 pg/mL vs. IQR75: >1.48 pg/mL) on combined endpoint (myocardial infarction, cardiovascular death, stroke/transient ischemic attack and death from stroke) in 10-year follow-up (IQR: interquartile range).

**Figure 5 biomolecules-14-00979-f005:**
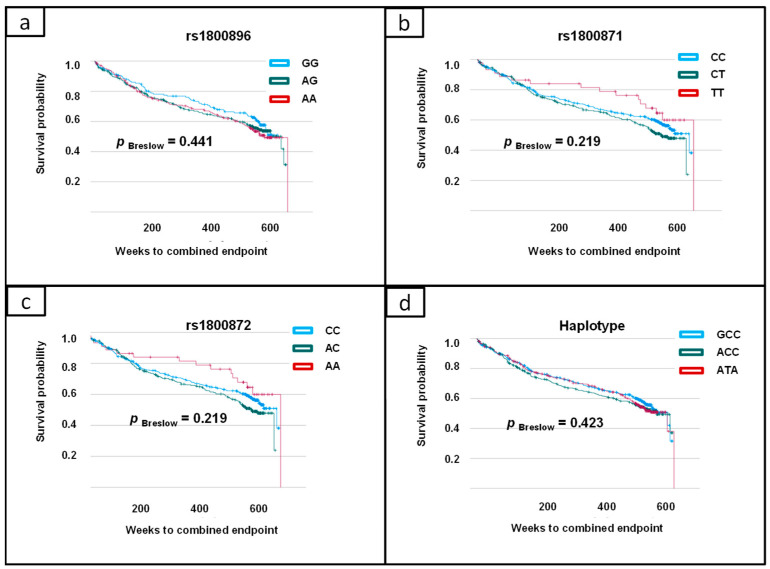
Kaplan–Meier survival curve including Breslow test: evaluation of genetic variants in IL-10 gene as prognostic factors for 10-year combined endpoint (myocardial infarction, cardiovascular death, stroke/transient ischemic attack and death from stroke): (**a**) rs1800896; (**b**) rs1800871; (**c**) rs1800872; (**d**) haplotypes of rs1800896/rs1800871/rs1800872.

**Table 1 biomolecules-14-00979-t001:** Cox regression for the multivariate assessment of the prognostic impact of an elevated IL-10 level (≥1.5 pg/mL) on adverse cardiovascular outcomes (myocardial infarction, cardiovascular death, stroke/transient ischemic attack and death from stroke) (CI: confidence interval; BMI: body mass index).

	Regression Coefficient	Standard Error	*p*-Value	Hazard Ratio	95% CI	
					Upper	Lower
IL-10 level ≥ 1.5 pg/mL	0.232	0.169	0.168	1.26	0.91	1.76
Age	0.037	0.010	<0.001	1.04	1.02	1.06
Male gender	0.350	0.200	0.080	1.42	0.96	2.10
BMI	−0.045	0.024	0.057	0.96	0.91	1.00
Smoking	0.120	0.362	0.740	1.13	0.56	2.29
Diabetes mellitus	0.385	0.179	0.032	1.47	1.03	2.09
Hypertension	0.002	0.278	0.995	1.00	0.58	1.73
Hyperlipoproteinemia	−0.138	0.173	0.426	0.87	0.62	1.22

## Data Availability

The original contributions presented in this study are included in the article. Further inquiries can be directed to the corresponding author.

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
