# Peer review of "Major Adverse Cardiovascular Events: The Importance of Serum Levels and Haplotypes of the Anti-Inflammatory Cytokine Interleukin 10"

_biomolecules, 2024, doi:10.3390/biom14080979_

Round 1

Reviewer 1 Report

Comments and Suggestions for Authors

Nice results and findings. The manuscript can be accepted in its present form.

Author Response

Dear Reviewer,

thank you very much for your valuable and favourable assessment of our work.

I would like to thank you on behalf of all the co-authors for your very positive feedback on our manuscript.

With kindest regards

Susanne Schulz on behalf of all authors

Reviewer 2 Report

Comments and Suggestions for Authors

By this manuscript, Authors wanted to assess IL-10 (Haplotype and SNPs) as a potential biomarker for CVD (10 yrs follow-up study).

Background paragraph correctly introduce the topic focusing on the study details which are fully described in Matherial and Methods section. Results are presented nicely and are very clear and support conclusion; statistical analysis done is complete and there are no concerns on this aspect

Considering that the IL-10 level for cardiovascular events is strongly dependent on the study design and that no specific SNP could be a prognostic cardiovascular factor and that IL-10 expression is dependent on genetic predisposition, all these points leading to the hypothesis that fluctuation of IL-10 shall be associated directly or indirectly on different factors which might or might not be associated with CVD. Hence, and as Authors indicated, IL-10 could be a suitable biomarker for CVD. Nevertheless, this manuscript results in a deep dive on the topic presenting and comparing results obtained in other clinical studies.

Minor concern:

Fig. 3: title of the picture reports IL-10 as well as figure legend however figure expaination refers to IL-6. I think it's a typo which needs to be corrected.

Author Response

Dear Reviewer,

Thank you very much for the favorable review of our work presented here.

Considering that the IL-10 level for cardiovascular events is strongly dependent on the study design and that no specific SNP could be a prognostic cardiovascular factor and that IL-10 expression is dependent
on genetic predisposition, all these points leading to the hypothesis that fluctuation of IL-10 shall be associated directly or indirectly on different factors which might or might not be associated with CVD.
Hence, and as Authors indicated, IL-10 could be a suitable biomarker for CVD. Nevertheless, this manuscript results in a deep dive on the topic presenting and comparing results obtained in other clinical studies.

The complex interaction of a large number of risk factors leading to the occurrence of further cardiovascular events was taken into account by means of the inclusion of established cardiovascular risk factors applying cox regression.

Fig. 3: title of the picture reports IL-10 as well as figure legend however figure expaination refers to IL-6. I think it's a typo which needs to be corrected.

Thank you for the valuable advice. The mistake has been corrected.

With kindest regards

Susanne Schulz on behalf of all authors        

Reviewer 3 Report

Comments and Suggestions for Authors

The concept of the study is intersting and the writing and data presentation are easy to follow. Nevertheless, the study presents some flaws that place under question the findings of the study. 

-The study has been conducted exclusively on patients with established CVD. The importance of IL-10 serum levels and haplotype analysis in these patients may be seriously questioned, as these patients are already characterized by very high CVD risk. 

- The study is limited by its design as a sub-study of a longitudinal cohort study designed for other reason

- Of the 1,002 patients who met the inclusion and exclusion criteria, more than half were excluded from protein analysis. This is a high percentage. The authors mention as reason for this exclusion the determination of IL-10 levels below measurement limits. Why was IL-10 in these samples below measurement limit? Are the authors justified to exclude these patients for this reason or is this biased? Baseline characteristics of the patients not included definetely need to be presented versus those who were included for protein analysis.

- Similarly, did patients who were excluded from genetic analysis differ from those who were included?

- To evaluate the influence of IL-10 serum levels on CVD outcome, IL-10 were dichotomized. Did the authors perform an analysis based on 25/75IQR? This would probably have been more meaningful, as upper and lower values of biomarkers are used for outcome prediction. 

- Did the authors evaluate levels of other interleuikins or other markers of subclinical inflammation such as hs-CRP?

- In the Abstract, the "conclusion" is not actually a conclusion of the study findings.

- The introduction needs to be slightly shortened, as some parts appear to fit better in the discussion section. 

- The discussion should start by summarizing the study findings.

Round 2

Reviewer 3 Report

Comments and Suggestions for Authors

The manuscript has been significantly improved in its revised version. No further comments.